# Physiological roles of endocytosis and presynaptic scaffold in vesicle replenishment at fast and slow central synapses

**Satyajit Mahapatra\*, Tomoyuki Takahashi\***

Cellular and Molecular Synaptic Function Unit, Okinawa Institute of Science and Technology - Graduate University, Okinawa, Japan

**\*For correspondence:**
satyajit.mahapatra@oist.jp (SM);
ttakahas@oist.jp (TT)

**Competing interest:** The authors declare that no competing interests exist.

**Abstract** After exocytosis, release sites are cleared of vesicular residues to replenish with transmitter-filled vesicles. Endocytic and scaffold proteins are thought to underlie this site-clearance mechanism. However, the physiological significance of this mechanism at diverse mammalian central synapses remains unknown. Here, we tested this in a physiologically optimized condition using action potential evoked EPSCs at fast calyx synapse and relatively slow hippocampal CA1 synapse, in post-hearing mice brain slices at 37°C and in 1.3 mM [$Ca^{2+}$]. Pharmacological block of endocytosis enhanced synaptic depression at the calyx synapse, whereas it attenuated synaptic facilitation at the hippocampal synapse. Block of scaffold protein activity likewise enhanced synaptic depression at the calyx but had no effect at the hippocampal synapse. At the fast calyx synapse, block of endocytosis or scaffold protein activity significantly enhanced synaptic depression as early as 10 ms after the stimulation onset. Unlike previous reports, neither endocytic blockers nor scaffold protein inhibitors prolonged the recovery from short-term depression. We conclude that the release-site clearance by endocytosis can be a universal phenomenon supporting vesicle replenishment at both fast and slow synapses, whereas the presynaptic scaffold mechanism likely plays a specialized role in vesicle replenishment predominantly at fast synapses.

## eLife assessment

Following synaptic vesicle fusion events at release sites, vesicle remnants will need to be cleared in order to allow new rounds of vesicle docking and fusion. This **fundamental** study of Mahapatra and Takahashi examines the role of release site clearance in synaptic transmission during repetitive activity in two types of central synapses, the giant calyx of Held and hippocampal CA1 synapses. The study uses pharmacological approaches to interfere with release site clearance by blocking membrane retrieval (endocytosis). The results also show how pharmacological inhibition of scaffold proteins affects short-term plasticity. The data presented make a **compelling** case for fast endocytosis as necessary for rapid site clearance and vesicle recruitment to active zones. The data reveal an unexpected, fast role for local site clearance in counteracting synaptic depression.

## Introduction

Chemical synaptic transmission depends upon fusion of transmitter-filled vesicles with the presynaptic membrane. At presynaptic terminals, there are a limited number of vesicular release sites, which can become refractory while discharged vesicles are remaining at the site (*Katz, 1993*), inhibiting subsequent vesicle fusion. Following inactivation of the temperature-sensitive endocytic protein dynamin in

*Drosophila* mutant *Shibire*, short-term depression (STD) of neuromuscular synaptic currents evoked by action potentials (APs) are enhanced within 20 ms from the stimulation onset at 33°C (*Kawasaki et al., 2000*), suggesting that endocytosis plays a significant role in rapid release site-clearance in addition to its well-established role in vesicle recycling (*Neher and Sakaba, 2008*). The enhancement of STD by blocking endocytosis was reproduced at the calyx of Held in slices from rodent brainstem, associated with a pronounced prolongation in the recovery from STD of EPSCs evoked by paired command pulses under voltage clamp (*Hosoi et al., 2009*). Fluorescence imaging with synaptopHluorin at cultured hippocampal synapses also showed that endocytic blockers, but not the vesicle acidification blockers, enhance steady state depression of exocytosis induced by high-frequency stimulation, suggesting slow clearance of vesicular component from release-sites (*Hua et al., 2013*). Also, in cell culture studies, genetic ablation of the endocytic adaptor protein AP-2μ (*Jung et al., 2015*), synaptophysin (*Rajappa et al., 2016*), or the secretory carrier membrane protein SCAMP5 (*Park et al., 2018*) enhances STD.

However, it remains open whether the endocytosis-dependent release site-clearance phenomenon reported at the *Drosophila* neuromuscular junction (*Kawasaki et al., 2000*) plays a physiological role at mammalian central synapses.

The presynaptic scaffold protein intersectin is a guanine nucleotide exchange factor, which activates the Rho-GTPase CDC42, thereby regulating the filamentous (F) actin assembly (*Hussain et al., 2001*; *Marie et al., 2004*). At the calyx of Held in slices from pre-hearing rodents, genetic ablation of intersectin 1 or pharmacological block of CDC42 abolishes the fast component of the rate of recovery from STD of EPSCs evoked by a pair of square pulses (*Sakaba et al., 2013*). They suggest that the scaffold protein cascade, comprising intersectin, CDC42, and F-actin, contributes to rapid vesicle replenishment, possibly through release site-clearance (*Jäpel et al., 2020*).

Like the endocytosis-dependent site-clearance mechanism, its physiological significance at diverse mammalian central synapses remains unknown. To address these issues, we evoked EPSCs by afferent fiber stimulations in post-hearing (P13-15) mice brain slices, without presynaptic intracellular perturbation, at brainstem calyces of Held and hippocampal CA1 synapses at physiological temperature (PT, 37°C; *Sanchez-Alavez et al., 2011*) and in artificial cerebrospinal fluid (aCSF) containing 1.3 mM $Ca^{2+}$, in reference to rodent CSF (*Jones and Keep, 1988*; *Silver and Erecińska, 1990*; *Inglebert et al., 2020*). The post-hearing calyx of Held is a fast-signaling auditory relay synapse that can respond to inputs up to 500 Hz without failure in vivo (*Sonntag et al., 2009*) and displays STD during repetitive stimulation. In contrast, the hippocampal synapse from Schaffer collaterals (SCs) to pyramidal cells (PCs) in the CA1 area is a relatively slow synapse that starts to fail transmission already at 40 Hz in slice (*Combe et al., 2018*) and at 37°C and in 1.3 mM [$Ca^{2+}$], short-term facilitation masks depression. At these synapses, we tested the effect of blocking endocytosis or scaffold protein cascade activity on EPSCs evoked by repetitive afferent fiber stimulations (30x) at different frequencies. These experiments indicated that the endocytosis-driven site-clearance mechanism activity-dependently supports vesicle replenishment, thereby counteracting synaptic depression at brainstem calyceal synapses and boosting synaptic facilitation at hippocampal CA1 synapses. In contrast, the vesicle replenishing role of the scaffold mechanism is activity and endocytosis independent and restricted to fast calyceal synapses, where it co-operates with the endocytosis-dependent site-clearance mechanism to upregulate synaptic strength, thereby maintaining the high-fidelity fast neurotransmission.

## Results

### Potency of endocytic blockers on slow and fast endocytosis at the calyx of Held

To clarify physiological roles of endocytosis in high-frequency synaptic transmission, we first examined the effect of different endocytic inhibitors on fast and slow endocytosis, using presynaptic membrane capacitance measurements (*Sun et al., 2004*; *Yamashita et al., 2010*) at post-hearing calyces of Held (P13-15) at PT (37°C; *Renden and von Gersdorff, 2007*). Since genetic ablation of endocytic proteins can be associated with robust compensatory effects (*Ferguson et al., 2009*; *Park et al., 2013*), we adopted acute pharmacological block using endocytic inhibitors Dynasore and Pitstop-2. For these experiments, we used 2.0 mM [$Ca^{2+}$] aCSF (*Figure 1*) to allow direct comparison with previous reports. Dynasore is a dynamin-dependent endocytosis blocker (*Macia et al., 2006*; *Newton et al.,*

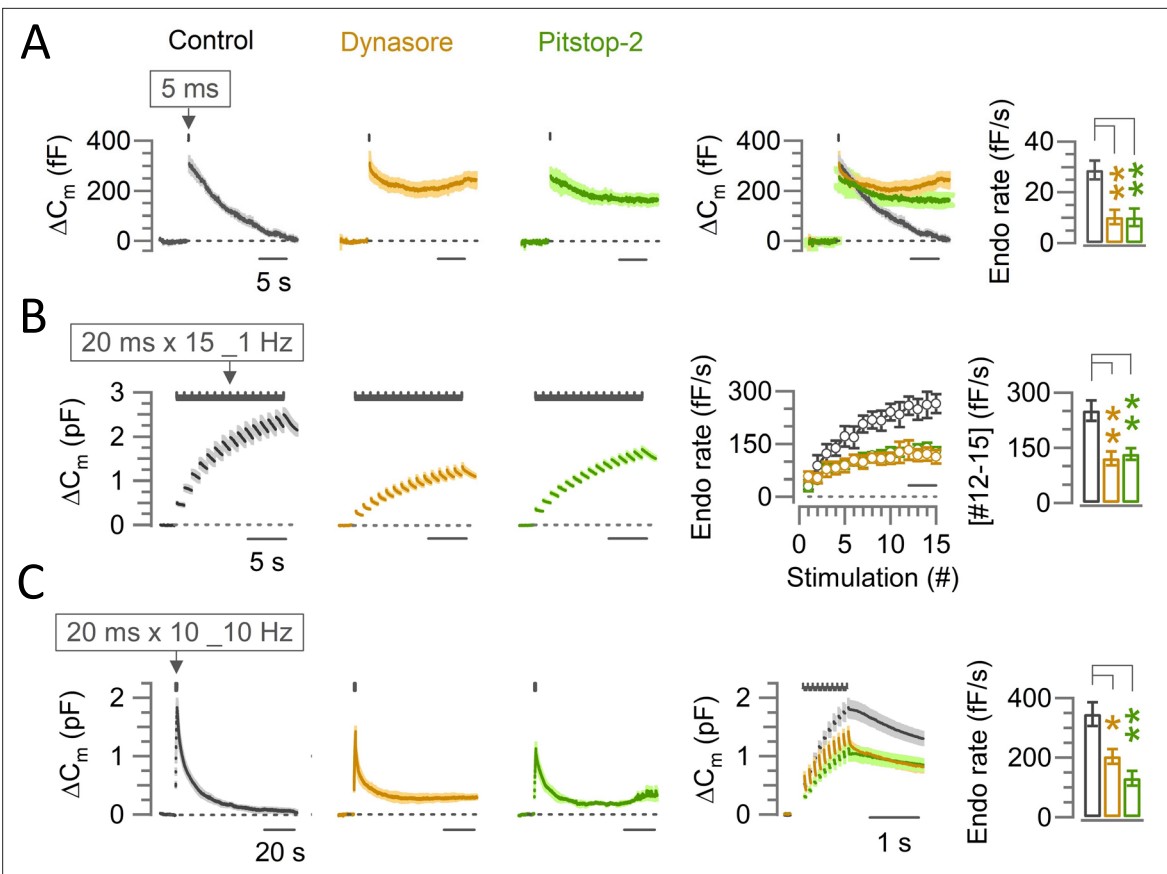

**Figure 1.** The endocytic blocker Dynasore or Pitstop-2 inhibits slow, fast-accelerating and fast endocytosis at the calyx of Held. (**A**) Average traces of slow endocytic membrane capacitance changes ($\Delta C_m$) in response to a 5 ms depolarizing pulse (stepping from –70 mV to +10 mV) in the absence (control, black trace) or presence of Dynasore (100 µM, 10–60 min, brown trace) or Pitstop-2 (25 µM, 10–60 min, green trace), recorded from the calyx of Held presynaptic terminal in slices from P13-15 post-hearing mice at physiological temperature (PT, 37°C) and in 2.0 mM $Ca^{2+}$ aCSF. The 4th panel from the left shows the superimposed average $\Delta C_m$ traces under control, Dynasore and Pitstop-2. The rightmost bar graph shows the endocytic decay rate (calculated from the slope 0.45–5.45 s after stimulation) that was slower in the presence of Dynasore (10.3±2.8 fF/s; n=5; p=0.004, Student's t-test) or Pitstop-2 (10.1±3.4 fF/s; n=5; p=0.006) than control (28.8±3.7 fF/s; n=5). (**B**) Fast-accelerating endocytosis induced by a train of 20 ms depolarizing pulses (repeated 15 times at 1 Hz) in the absence (control) or presence of Dynasore or Pitstop-2. Averaged traces are shown as in (A). The 4th panel shows the endocytic rates (fF/s) calculated from the slope of $C_m$ decay, 0.45–0.95 ms after each stimulation pulse under control, Dynasore, or Pitstop-2 (superimposed). The rightmost bar graph shows the endocytic rate averaged from stimulations #12–15 (bar in 4th panel) that was slower in the presence of Dynasore (121±19 fF/s; n=6; p=0.0045, Student's t-test) or Pitstop-2 (133±16 fF/s; n=5; p=0.007) than control (251±28 fF/s; n=6), indicating significant inhibition of the fast-accelerating endocytosis by Dynasore or Pitstop-2 (also see ***Supplementary file 1***). (**C**) Fast-endocytosis (average traces) evoked by a train of 20 ms pulses (repeated 10 times at 10 Hz) in the absence (control) or presence of Dynasore or Pitstop-2. The 4th panel shows cumulative $\Delta C_m$ traces (superimposed) in an expanded timescale during and immediately after the 10 Hz stimulation train. The rightmost bar graph indicates endocytic decay rates (measured 0.45–1.45 s after the 10th stimulation) in Dynasore (204±25.3 fF; n=4; p=0.017, Student's t-test) or in Pitstop-2 (131±24.6 fF; n=4; p=0.004) both of which were significantly slower than control (346±40.1 fF; n=6). All data in this figure are presented as mean ± sem and the statistical significance level was set at p<0.05, denoted with asterisks (*p<0.05, **p<0.01, ***p<0.001). The significance of effect was evaluated using one-way ANOVA and Student's t-test, with Bonferroni-Holm method of p level correction.

The online version of this article includes the following source data and figure supplement(s) for figure 1:

**Source data 1.** The endocytic blocker Dynasore or Pitstop-2 inhibits slow, fast-accelerating and fast endocytosis at the calyx of Held.

**Figure supplement 1.** Effect of intra-terminal loading of Dynamin-1 PRD peptide on endocytosis at the calyx of Held.

**Figure supplement 1—source data 1.** Effect of intra-terminal loading of Dynamin-1 PRD peptide on endocytosis at the calyx of Held.

*2006*) blocking both clathrin-mediated and clathrin-independent endocytosis (***Park et al., 2013***; ***Delvendahl et al., 2016***). A second endocytosis blocker Pitstop-2 is thought to preferentially block clathrin-mediated endocytosis (***von Kleist et al., 2011***; ***Delvendahl et al., 2016***; ***López-Hernández et al., 2022***) but it reportedly blocks clathrin-independent endocytosis as well (***Dutta et al., 2012***; ***Willox et al., 2014***).

At calyces of Held, a single short (5 ms) command pulse elicited an exocytic capacitance jump ($\Delta C_m$) followed by a slow endocytosis (29 fF/s in decay rate, *Figure 1A*). In the presence of Dynasore (100 μM) or Pitstop-2 (25 μM) in the perfusate (within 10–60 min of application), this slow endocytic membrane recovery was almost completely blocked, whereas $\Delta C_m$ or presynaptic $Ca^{2+}$ current was unchanged (*Supplementary file 1*). In a repetitive stimulation protocol, the accelerating endocytosis can be induced by a 1 Hz train of 20 ms square pulses (*Wu et al., 2005*; *Yamashita et al., 2010*). Dynasore or Pitstop-2 significantly inhibited the maximal rate of 250 fF/s of this accelerated fast endocytosis (20 ms x 15 times at 1 Hz) to ~50% (*Figure 1B*, *Supplementary file 1*). In another stimulation protocol (*Figure 1C*), 10 command pulses (20 ms duration) applied at 10 Hz induced a fast endocytosis with a decay rate (350 fF/s) by more than 10 times faster than the slow endocytosis elicited by a 5 ms pulse (*Figure 1A*). Dynasore or Pitstop-2 significantly inhibited the rate of this fast endocytosis (*Figure 1C*, *Supplementary file 1*) too. The potencies of bath-applied Dynasore or Pitstop-2 for blocking slow and fast forms of endocytosis were comparable to that of dynamin-1 proline-rich domain peptide (Dyn-1 PRD peptide) directly loaded in calyceal terminals (1 mM; *Figure 1—figure supplement 1*; *Yamashita et al., 2005*). Thus, at calyces of Held, bath-application of Dynasore or Pitstop-2 can block both fast and slow endocytosis to a similar extent as direct presynaptic whole-cell loading of Dyn-1 PRD peptide without perturbing presynaptic intracellular milieu.

## Effects of endocytic blockers on synaptic depression and recovery from depression at brainstem calyceal synapses

Using Dynasore or Pitstop-2, we investigated the effect of blocking endocytosis on synaptic transmission at the calyx of Held in brainstem slices from post-hearing mice (P13-15). At PT (37°C) in physiological aCSF (1.3 mM [$Ca^{2+}$]), EPSCs were evoked by a train of afferent fiber stimulations (30x at 10 and 100 Hz; *Figure 2*). Neither Dynasore nor Pitstop-2 (in the perfusate, within 10–60 min of application) affected the first EPSC amplitude in the train (*Figure 2—figure supplement 1A*). During a train of stimulations at 10 Hz, in the presence of Dynasore (100 μM), synaptic depression was marginally enhanced (from 45% to 52%, with no significant difference), but Pitstop-2 had no effect (*Figure 2A1*). At 100 Hz stimulations, however, both Dynasore and Pitstop-2 markedly and equally enhanced synaptic depression starting at 10 ms (2nd stimulation) from the stimulus onset, increasing the magnitude of steady state STD (averaged from #26–30 EPSCs) from 58% to 75% (1.3 times; $p<0.001$, t-test; *Figure 2B1*). These results suggest that endocytosis-dependent rapid SV replenishment operates during high-frequency transmission at the mammalian central synapses like at the neuromuscular junction (NMJ) of *Drosophila* (*Kawasaki et al., 2000*). Since glutamate released during action potential evoked EPSCs does not desensitize or saturate postsynaptic receptors at post-hearing calyces of Held (*Ishikawa et al., 2002*) unlike at pre-hearing calyces (*Yamashita et al., 2009*), enhanced synaptic depression in the presence of endocytic blockers during repetitive fiber stimulations is most likely caused by presynaptic mechanisms. In fact, the results of the endocytic block were essentially the same in the absence (*Figure 2*) or presence (*Figure 2—figure supplement 2*) of the low-affinity glutamate receptor ligand kynurenic acid (1 mM), which reduces postsynaptic receptor occupancy with glutamate.

At pre-hearing calyces of Held at room temperature (RT) and in 2.0 mM [$Ca^{2+}$], the recovery from STD of EPSCs evoked by paired command pulse (50 ms) stimulations was markedly prolonged by endocytic inhibition (*Hosoi et al., 2009*). In contrast, at post-hearing calyces in 1.3 mM [$Ca^{2+}$] and at PT (37°C), the endocytic blockers did not prolong the recovery of AP evoked EPSCs from STD (*Figure 2A2 and B2*, *Supplementary file 2*). Conversely, the recovery from STD caused by 100 Hz stimulation was significantly accelerated at both fast and slow recovery components (*Figure 2B2*, *Supplementary file 2*). To explore the reason for these different results, we raised $Ca^{2+}$ concentration in aCSF to 2.0 mM at PT (*Figure 2—figure supplement 3A*), RT (*Figure 2—figure supplement 3B*) or raised stimulation frequency to 200 Hz at PT (*Figure 2—figure supplement 3C*). However, none of these manipulations caused significant change in the kinetics of recovery from STD (*Figure 2—figure supplement 3*). Remaining differences between previous and present results include the species- and/or age-differences of animals (pre-hearing rats *vs* post-hearing mice), with or without whole-cell perturbation, and strength of stimulation intensity, which is stronger in the voltage-clamp protocol with paired command pulses (50 ms) than APs. The reason for the slight recovery acceleration in our protocol remains unknown. Since the acceleration was absent in 2.0 mM [$Ca^{2+}$] (*Figure 2—figure supplement 3*) in the presence of Dynasore or Pitstop-2 as previously observed in dynamin-1 knockout

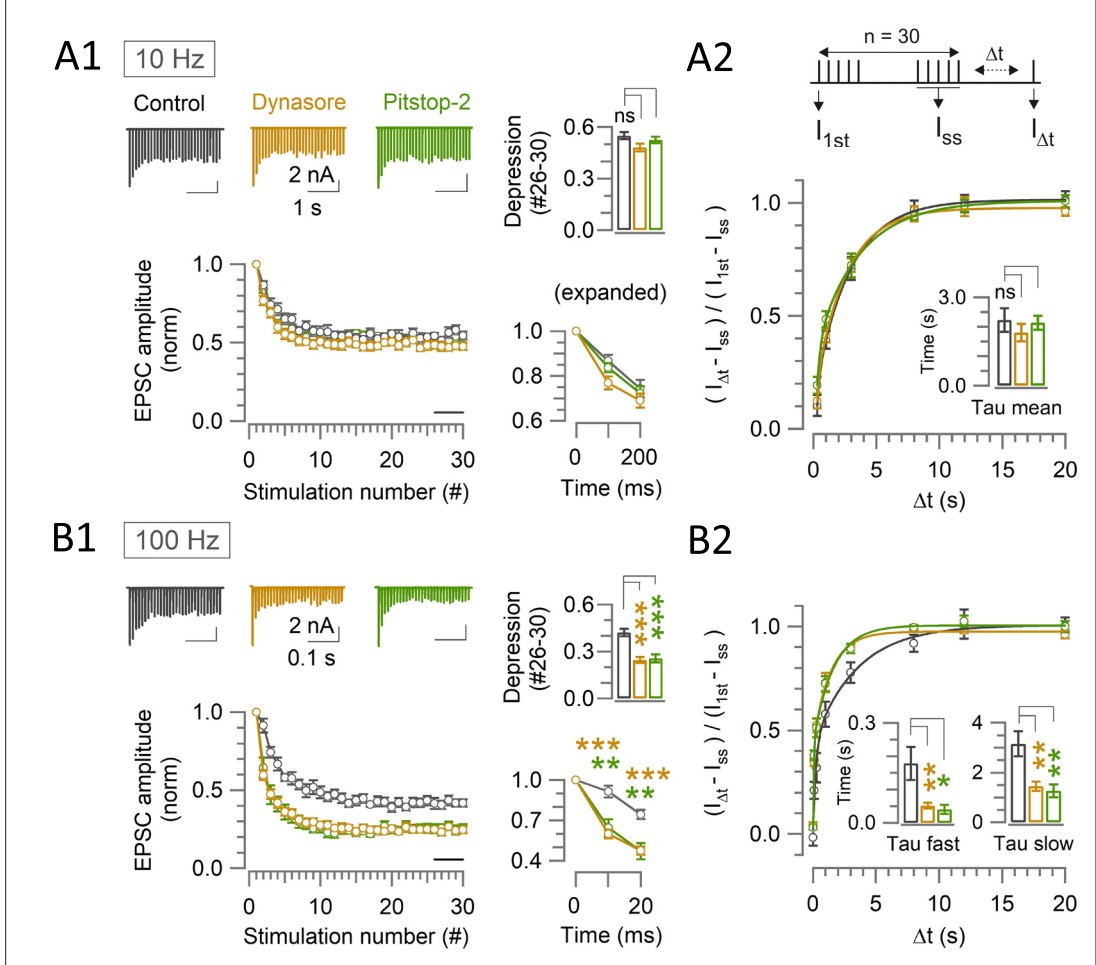

**Figure 2.** Endocytic blockers enhance activity dependent rapid synaptic depression, but do not prolong the recovery from depression at the calyx. (**A1, B1**) A train of 30 EPSCs were evoked at the calyx of Held by afferent fiber stimulation at 10 Hz (**A1**) or 100 Hz (**B1**) in the absence (control, black traces) or presence of Dynasore (10–60 min, brown traces) or Pitstop-2 (10–60 min, green traces) at PT (37°C) in 1.3 mM $Ca^{2+}$ aCSF. *Panels from left to right:* left-top: sample EPSC traces; left-bottom: normalized average EPSC amplitudes at each stimulus number; right-top bar graph: steady-state depression of EPSC amplitudes (mean of EPSCs #26–30, bar in the second panel); right-bottom: normalized 1st - 3rd EPSC amplitudes in expanded timescale. (**A2, B2**) The recovery of EPSCs from STD in control, or in the presence of Dynasore or Pitstop-2 at the calyx of Held measured using a stimulation protocol (shown on top in A2); a train of 30 stimulations at 10 Hz (**A2**) or 100 Hz (**B2**) followed by test pulses after different time intervals (Δt: 0.02, 0.1, 0.3, 1, 3, 8, 12, and 20 s). The EPSC amplitude after Δt ($I_{\Delta t}$) relative to the first EPSC in the stimulus train ($I_{1st}$) was normalized by subtracting the steady state EPSCs ($I_{ss}$) to measure the recovery rates. (**A1**) During 10 Hz stimulation, the steady-state depression under control (0.55±0.02; n=11) was unaltered in the presence of either Dynasore (0.48±0.02; n=12; p=0.03, not significantly different, One-way ANOVA and Student's t-test with Bonferroni-Holm correction) or Pitstop-2 (0.53±0.02; n=7; p=0.4, no significant difference). (**A2**) After 10 Hz stimulation, the time constant of EPSCs recovery in control (2.3±0.4 s; n=9) was unchanged in the presence of either Dynasore (1.7±0.2 s; n=8; p=0.2, Student's t-test) or Pitstop-2 (1.9±0.3 s; n=7; p=0.4). (**B1**) During 100 Hz stimulation, EPSCs underwent a significant depression starting at the 2nd stimulation (10 ms) in the presence of both Dynasore (0.6±0.03; n=10; p<0.001, t-test) or Pitstop-2 (0.65±0.06; n=7; p=0.003) than control (0.91±0.05; n=11). Bar graph indicates steady-state STD magnitudes; control: (0.42±0.025), Dynasore: (0.25±0.02; p<0.001), and Pitstop-2: (0.26±0.03; p<0.001). (**B2**) After 100 Hz stimulation, both fast and slow recovery time constants were significantly faster in the presence of Dynasore ($\tau_{fast}$: 0.05±0.009 s; n=8; p=0.008 and $\tau_{slow}$: 1.5±0.2 s; p=0.003) or Pitstop-2 ($\tau_{fast}$: 0.04±0.014 s; n=6; p=0.02 and $\tau_{slow}$: 1.3±0.3 s; p=0.008) than control ($\tau_{fast}$: 0.18±0.05 s; n=8 and $\tau_{slow}$: 3.2±0.5 s) ***Supplementary file 2***. The online version of this article includes the following source data and figure supplement(s) for ***Figure 2***:

The online version of this article includes the following source data and figure supplement(s) for figure 2:

**Source data 1.** Endocytic blockers enhance activity dependent rapid synaptic depression, but do not prolong the recovery from depression at the calyx.

**Figure supplement 1.** Effect of endocytic inhibitors or scaffold inhibitors on basal EPSC amplitude at the calyx and hippocampal CA1 synapses.

**Figure supplement 2.** Postsynaptic AMPA receptor saturation does not contribute to enhancement of synaptic depression by endocytic blockers at the calyx of Held.

**Figure supplement 2—source data 1.** Postsynaptic AMPA receptor saturation does not contribute to enhancement of synaptic depression by

*Figure 2 continued on next page*

*Figure 2 continued*

endocytic blockers at the calyx of Held.

**Figure supplement 3.** Endocytic blockers enhance STD but do not alter the recovery from STD time-course in 2.0 mM [Ca²⁺] irrespective of PT or RT at the calyx of Held.

**Figure supplement 3—source data 1.** Endocytic blockers enhance STD but do not alter the recovery from STD time-course in 2.0 mM [Ca²⁺] irrespective of PT or RT at the calyx of Held.

mice (*Mahapatra et al., 2016*), there might be a low [$Ca^{2+}$]-dependent recovery acceleration mechanism distinct from that enhanced by high [$Ca^{2+}$] (CDR; *Wang and Kaczmarek, 1998*).

## Effects of scaffold protein cascade inhibitors on vesicle endocytosis, synaptic depression, and recovery from depression at brainstem calyceal synapses

Among presynaptic scaffold proteins, F-actin is assembled by the Rho-GTPase CDC42 activated by the scaffold protein intersectin, a guanine nucleotide exchange factor (*Hussain et al., 2001*; *Marie et al., 2004*). Genetic ablation of intersectin 1 or pharmacological inhibition of CDC42 activity significantly prolongs the recovery from STD at pre-hearing calyces of Held, suggesting that this scaffold cascade contributes to vesicle replenishment, possibly via release site-clearance (*Sakaba et al., 2013*; *Jäpel et al., 2020*).

Genetic ablation of intersectin inhibits endocytosis in cultured synapses (*Yu et al., 2008*), but does not affect endocytosis at pre-hearing calyces of Held (*Sakaba et al., 2013*). We examined the effect of blocking CDC42 using ML141 (10 µM, 10–60 min) or inhibiting F-actin assembly using Latrunculin-B (15 µM, 10–60 min) in perfusates on fast and slow endocytosis at the post-hearing (P13-15) calyx of Held at PT (*Figure 3*) in aCSF containing 2.0 mM [$Ca^{2+}$]. In agreement with previous report (*Sakaba et al., 2013*), none of these inhibitors affected the fast or slow forms of endocytosis.

We then recorded EPSCs in 1.3 mM [$Ca^{2+}$] at PT. Neither ML141 nor Latrunculin-B affected the first EPSC amplitude in the train (*Figure 2—figure supplement 1A*), alike Latrunculin-B effects at RT and 2.0 mM [$Ca^{2+}$] (*Mahapatra et al., 2016*; *Mahapatra and Lou, 2017*). During a train of stimulations at 10 Hz (*Figure 4A1*) or 100 Hz (*Figure 4B1*), in the presence of ML141 or Latrunculin-B in perfusates, synaptic depression was markedly and equally enhanced, with STD magnitude (#26–30) increasing from 45% to 62% (1.4 times; *P*<0.001, t-test; *Figure 4A1*) at 10 Hz and from 58% to 80% at 100 Hz (1.4 times; p<0.001, t-test; *Figure 4B1*). Thus, in contrast to endocytic blockers (*Figure 2*), the depression-enhancing effects of scaffold cascade inhibitors were independent of activity and endocytosis.

Like endocytic blockers, scaffold cascade inhibitors did not prolong the recovery from STD at 10 Hz (*Figure 4A2*) or at 100 Hz (*Figure 4B2*). These results contrast with those previously reported at pre-hearing calyces (*Sakaba et al., 2013*), where genetic ablation of intersectin or pharmacological block of CDC42 prolongs the recovery from STD induced by the paired command pulse (50 ms) stimulations under voltage-clamp.

The F-actin depolymerizers Latrunculin A and Latrunculin B are widely utilized to examine presynaptic roles of F-actin. However, the results of STD and recovery from STD can vary according to the application methods and experimental conditions (*Figure 4—figure supplement 1A*). At the calyx of Held, in 2.0 mM [$Ca^{2+}$] aCSF at PT, Latrunculin A perfusion (20 µM, 1 hr, *Figure 4—figure supplement 1A1*) showed a significant enhancement of STD at 100 Hz within 10–60 min of drug application (*Figure 4—figure supplement 1B1 and D*), as previously reported for Latrunculin B perfusion at RT and 200 Hz (*Mahapatra et al., 2016*). However, 1 hr preincubation of slices with Latrunculin A (20 µM) followed by perfusion with standard aCSF (~1 hr, *Figure 4—figure supplement 1A2*) had no significant effect on STD induced by a train of 30 stimuli at 100 Hz (*Figure 4—figure supplement 1B2 and D*), as previously reported (*Piriya Ananda Babu et al., 2020*). In 1.3 mM [$Ca^{2+}$] aCSF at PT (*Figure 4—figure supplement 1C*), marked STD enhancement was observed by continuous perfusion of either Latrunculin B (15 µM, *Figure 4—figure supplement 1C and D*) or Latrunculin A (20 µM, *Figure 4—figure supplement 1C and D*). Thus, Latrunculin A (20 µM) and Latrunculin B (15 µM) are equipotent for STD enhancement in these experiments (*Figure 4—figure supplement 1C and D*).

When the data are compared between 1.3 mM [$Ca^{2+}$] and 2.0 mM [$Ca^{2+}$], STD levels after Latrunculin A perfusion are not different (p>0.05, *Figure 4—figure supplement 1D*), whereas control STD

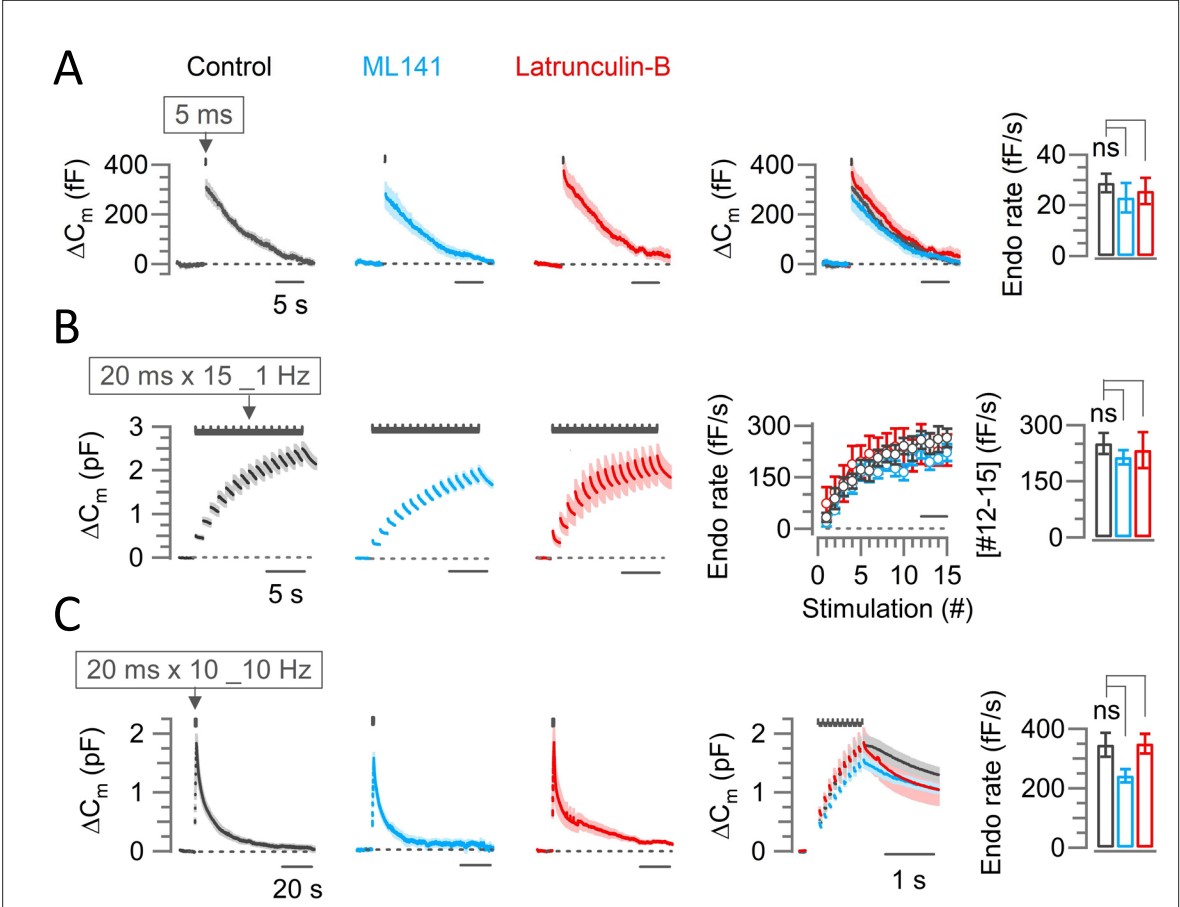

**Figure 3.** Scaffold machinery inhibitors have no effect on endocytic membrane retrievals at the calyx of Held. The CDC42 inhibitor ML141 (10 μM, 10–60 min, cyan) or actin depolymerizer Latrunculin B (Lat-B, 10–60 min, 15 μM, red) had no effect on slow endocytosis in response to a 5 ms depolarizing pulse (**A**) or on fast accelerating endocytosis (**B**; induced by 1 Hz train of 20 ms x 15 pulses) or fast endocytosis (**C**; evoked by a 10 Hz train of 20 ms x 10 pulses) at PT (37°C) and in 2.0 mM $Ca^{2+}$ aCSF at the post hearing-calyceal presynaptic terminals (P13-15) in slices. (**A**) Averaged and superimposed $\Delta C_m$ traces in response to a 5 ms pulse. The rightmost bar graph indicates the endocytic decay rate in control (28.8±3.7 fF; n=5), unchanged by ML141 (23.0±5.8 fF; n=6; p=0.44, Student's t-test) or Lat-B (25.6±5.2 fF; n=6; p=0.65; *Supplementary file 1*). (**B**) The average endocytic rate (#12–15) in control (251±28 fF/s; n=6) unaltered by ML141 (214±19 fF/s; n=5; p=0.3, Student's t-test) or Latrunculin-B (233±48 fF/s; n=4; p=0.7; *Supplementary file 1*). (**C**) Fast endocytic decay rate in the presence of ML141 (242±21.5 fF; n=4; p=0.08, t-test) or Lat-B (350±33.3 fF; n=5; p=0.95) was not different from control (346±40.1 fF; n=6; *Supplementary file 1*). Statistical significance of all data in this figure were tested using one-way ANOVA and Student's t-test, with Bonferroni-Holm method of p-level correction.

The online version of this article includes the following source data for figure 3:

**Source data 1.** Scaffold machinery inhibitors have no effect on endocytic membrane retrievals at the calyx of Held.

level in 1.3 mM [$Ca^{2+}$] (~0.4) was significantly less than 2.0 mM [$Ca^{2+}$] (~0.3; p=0.002, t-test; *Figure 4—figure supplement 1D*). Thus, physiological 1.3 mM [$Ca^{2+}$] revealed a robust STD-counteracting function of F-actin.

Regarding the recovery from STD, $Ca^{2+}$-dependent recovery (CDR) component of tens of milliseconds time constant is produced by strong stimulation at the calyx of Held at RT (*Wang and Kaczmarek, 1998*) and inhibited by Latrunculin B (*Lee et al., 2013*). CDR is also observed after STD induced by 100 Hz stimulation at PT and blocked by Latrunculin A pretreatment (20 μM) in 2.0 mM [$Ca^{2+}$] (*Piriya Ananda Babu et al., 2020*). However, such Latrunculin-sensitive fast recovery component was absent in 1.3 mM [$Ca^{2+}$] (*Figure 4B2*, *Supplementary file 2*). Thus, CDR may operate only when excessive $Ca^{2+}$ enters during massive presynaptic activation.

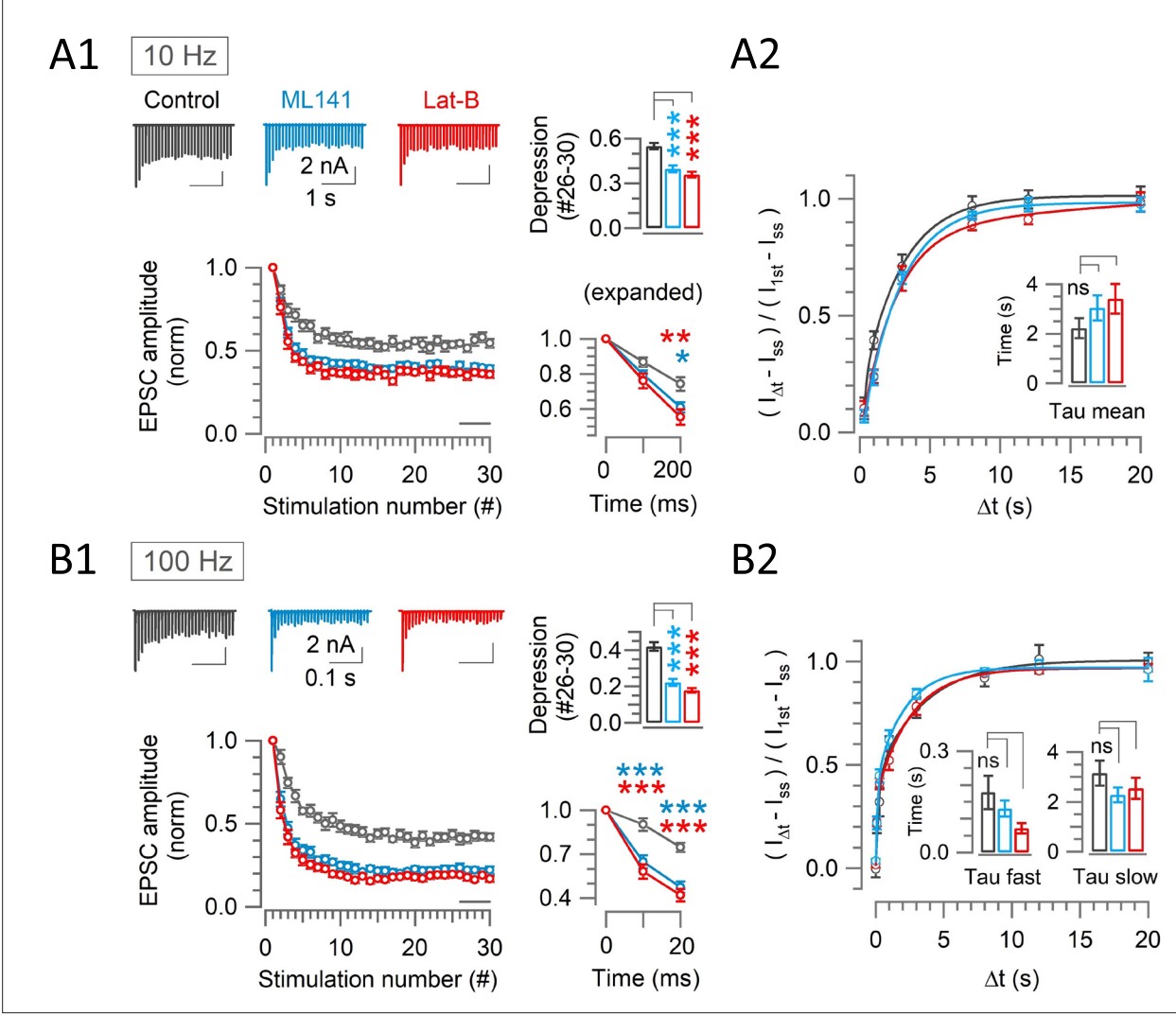

**Figure 4.** Scaffold machinery inhibitors strongly enhance rapid synaptic depression activity independently, without prolonging the recovery from depression at the calyx. (**A1, B1**) EPSCs (30x) evoked at the calyx of Held by afferent fiber stimulation at 10 Hz (**A1**) or 100 Hz (**B1**) in the absence (control, black) or presence of ML141 (10–60 min, cyan) or Lat-B (10–60 min, red) at PT and in 1.3 mM Ca$^{2+}$ aCSF. *Panels from left to right: like **Figure 2**.* (**A2, B2**) The recovery of EPSCs from STD in control, or in the presence of ML141 or Latrunculin-B at the calyx of Held measured at 10 Hz (**A2**) or 100 Hz (**B2**) – like ***Figure 2***. (**A1**) At 10 Hz stimulation, enhancement of depression became significant starting from the 3rd stimulation (200 ms) in the presence of ML141 (0.61±0.03; n=9; p=0.016, One-way ANOVA and Student's t-test with Bonferroni-Holm p level correction) or Lat-B (0.55±0.04; n=7; p=0.006) compared to control (0.74±0.04; n=11). Bar graph indicates the steady-state depression (STD) strongly enhanced from control (0.55±0.02) by ML141 (0.4±0.02; p<0.001) or Lat-B: (0.36±0.02; p<0.001). (**A2**) After a train of 30 stimulations at 10 Hz, the time constant of EPSCs recovery under control (2.2±0.4 s; n=9) was unchanged by ML141 (3.0±0.5 s; n=8; p=0.21) or Latrunculin-B (3.4±0.6 s; n=6; p=0.16; ***Supplementary file 2***). (**B1**) At 100 Hz stimulation, EPSCs showed significant enhancement of depression starting at the 2nd stimulation (10 ms) in the presence of ML141 (0.65±0.05; n=10; p<0.001) or Lat-B (0.58±0.05; n=8; p<0.001) from control (0.91±0.05; n=11). Bar graph indicates strong steady-state STD produced by ML141 (0.22±0.02; p<0.001) or Lat-B (0.18±0.013; p<0.001) compared to control (0.42±0.025). (**B2**) The time course of EPSC recovery from STD induced by a train of 30 stimulations at 100 Hz, indicating no significant change in recovery from STD time constants caused by ML141 ($\tau_{fast}$: 0.13±0.02 s; n=8; p=0.4 and $\tau_{slow}$: 2.3±0.3 s; p=0.15) or Lat-B ($\tau_{fast}$: 0.072±0.02 s; n=8; p=0.05 and $\tau_{slow}$: 2.5±0.4 s; p=0.4) from control ($\tau_{fast}$: 0.18±0.05 s; n=8 and $\tau_{slow}$: 3.2±0.5 s; ***Supplementary file 2***). One-way ANOVA and Student's t-test with Bonferroni-Holm p level correction was used to evaluate the statistical significance. The stimulation and recovery protocols were the same as those used in ***Figure 2***.

The online version of this article includes the following source data and figure supplement(s) for figure 4:

**Source data 1.** Scaffold machinery inhibitors strongly enhance rapid synaptic depression activity independently, without prolonging the recovery from depression at the calyx.

**Figure supplement 1.** Effects of Latrunculin application protocols and [Ca$^{2+}$] on STD.

**Figure supplement 1—source data 1.** Effects of Latrunculin application protocols and [Ca$^{2+}$] on STD.

## Kinetic comparison of synaptic depression enhanced by endocytic blockers and scaffold cascade inhibitors and evaluation of their combined effect

During 100 Hz stimulation, enhancement of synaptic depression by endocytic blockers or scaffold cascade inhibitors are prominent already in the second EPSCs (*Figures 2B1 and 4B1*). Exponential curve fits to the data points of STD time-course (*Figure 5A*) indicated that the EPSCs underwent a single exponential depression in controls with a mean time constant of 37 ms (n=11), whereas in the presence of endocytic blockers or scaffold cascade inhibitors, EPSCs amplitude underwent double exponential decay with a fast time constant of 5–10 ms and a slow time constant of 24–40 ms, the latter of which was similar to the time constant in control (*Figure 5B*). These data indicate that endocytic and scaffold mechanisms for vesicle replenishment operate predominantly at the beginning of synaptic depression. This is consistent with a lack of prolongation in the recovery from STD by endocytic blockers or scaffold cascade inhibitors (*Figure 2A2 and B2*; *Figure 4A2 and B2*).

Since the enhancement of synaptic depression by endocytic blockers or scaffold cascade inhibitors occurred mostly at the early phase of synaptic depression, endocytosis- and scaffold-dependent synaptic strengthening likely operate simultaneously during high-frequency transmission. To clarify whether these mechanisms are additive or complementary, we co-applied Dynasore and Latrunculin-B during 100 Hz stimulation (*Figure 5C*). The magnitude of EPSC depression by the co-application was the same as that by single Latrunculin-B application (*Figure 5C*), suggesting that endocytosis and scaffold mechanisms complementarily maintain synaptic strength during high-frequency transmission.

## Possible role of endocytosis and scaffold machineries at hippocampal synapses

Counteraction of synaptic depression by endocytosis or scaffold mechanism is highlighted at fast synapses of large structure such as neuromuscular junction or the calyx of Held. However, it remains open whether these mechanisms are conserved at other type of synapses. Hence, we recorded EPSCs from bouton-type hippocampal synapses at CA1 pyramidal neurons evoked by SC stimulation at 10 Hz or 25 Hz at 37°C and in aCSF containing 1.3 mM $[Ca^{2+}]$ (*Figure 6*). In this physiologically optimized experimental condition, EPSCs underwent a prominent facilitation and reached a peak at the 7th stimulation at 10 Hz and at the 12th stimulation at 25 Hz (*Figure 6*). In the presence of Dynasore (100 µM, within 10–60 min of application in perfusate), synaptic facilitation was much less than control, which was noticeable within 500 ms (6th stimulation) at 10 Hz stimulation or within 80 ms (3rd stimulation) at 25 Hz stimulation from the onset (*Figure 6A and B*). Unexpectedly, Dynasore significantly enhanced the basal EPSC amplitude (*Figure 2—figure supplement 1B*), whereas Pitstop-2 had no such effect (*Figure 2—figure supplement 1B*). Although less potent than Dynasore, Pitstop-2 significantly attenuated synaptic facilitation (*Figure 6A and B*). Thus, at these relatively slow bouton-type hippocampal synapses, endocytosis enhanced synaptic strength like at fast calyceal synapses to boost short-term synaptic facilitation.

Strikingly, unlike endocytic blockers, neither ML141 nor Latrunculin-B affected the hippocampal short-term facilitation during stimulations at 10 Hz or 25 Hz (*Figure 6C and D*). Thus, the presynaptic scaffold cascade proteins F-actin or CDC42 unlikely plays a regulatory role on short-term synaptic plasticity at hippocampal synapses. At the hippocampal CA1 synapse in slices at RT, block of vesicle acidification is reported to enhance synaptic depression at 10–30 Hz stimulation possibly due to a rapid recruitment of unfilled vesicles (*Ertunc et al., 2007*), whereas the same treatment in hippocampal culture shows no such effect (*Hua et al., 2013*). To determine whether the vesicle acidification blockers might enhance STD like endocytic blockers in physiologically optimized condition, we tested the effect of Bafilomycin and Folimycin on hippocampal EPSCs. Unlike in 2.0 mM $[Ca^{2+}]$ aCSF at RT (*Ertunc et al., 2007*), in 1.3 mM $[Ca^{2+}]$ aCSF at PT, EPSCs showed a prominent facilitation (*Figure 6*, *Figure 6—figure supplement 1*), and neither Folimycin (67 nM) nor Bafilomycin A1 (5 µM) significantly affected the short-term facilitation (*Figure 6—figure supplement 1*). Thus, physiologically, the vesicle acidification mechanism is not involved in the short-term synaptic efficacy at hippocampal CA1 synapses.

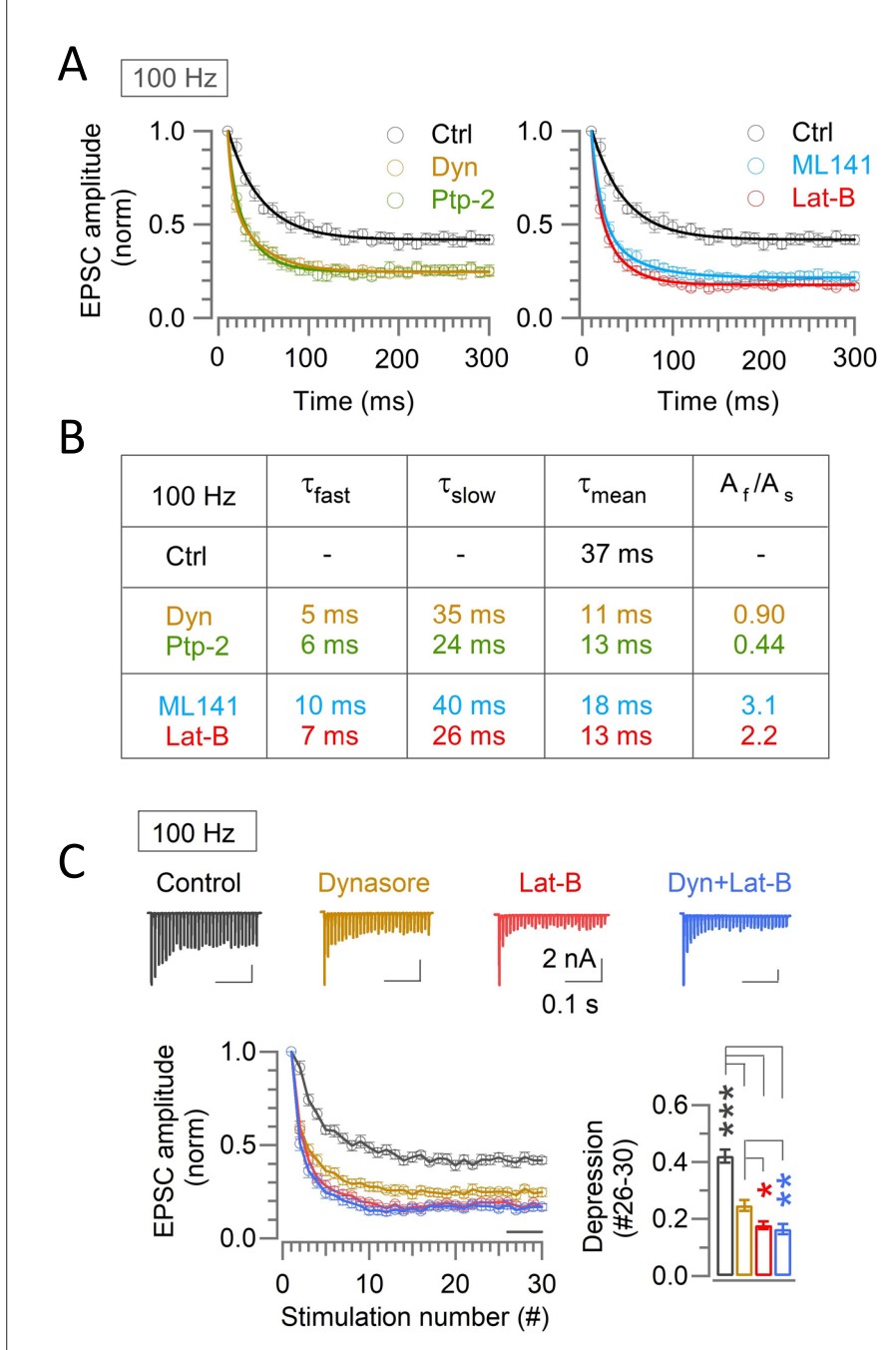

**Figure 5.** Endocytic and scaffold machineries co-operate for rapid vesicle replenishment during high-frequency transmission at the calyx of Held. (**A**) Exponential curve fits to the time-course of synaptic depression during 100 Hz stimulation under control and in the presence of either endocytic blockers or scaffold cascade inhibitors. The control time-course was best fit to a single exponential, whereas the time-course in the presence of endocytic blockers or scaffold cascade inhibitors was fit best to double exponential function with fast and slow time constants. (**B**) Parameters for the curve-fit, including fast and slow time constants ($\tau_{fast}$, $\tau_{slow}$), weighted mean time constant ($\tau_{mean}$) and relative ratio of fast and slow components ($A_f/A_s$). Similar fast time constant ($\tau_{fast}$) in the presence of either endocytic inhibitors or scaffold cascade inhibitors suggests simultaneous operation of both endocytic and scaffold mechanisms for countering the synaptic depression. Similar slow decay time constant irrespective of the presence or absence of blockers suggests that the endocytosis and scaffold mechanisms for vesicle replenishment operates predominantly at the beginning of the high frequency stimulations. (**C**) Enhancement of synaptic depression by co-application of Dynasore and Latrunculin-B (10–60 min) was like

*Figure 5 continued on next page*

*Figure 5 continued*

Latrunculin-B alone and stronger than Dynasore alone. Sample EPSC traces and normalized depression time courses are shown on the upper and lower left panels, respectively. The steady-state STD magnitudes are compared in bar graph; control: (0.42±0.025; n=11), Dynasore: (0.25±0.02; n=10; p<0.001 vs control; p=0.012 vs Lat-B; p=0.005 vs Dyn +Lat B), Latrunculin-B: (0.18±0.013; n=8; p<0.001 vs control; p=0.6 vs Dyn +Lat B), and Dyn +Lat B together: (0.17±0.02; n=11; p<0.001 vs control). Significance of difference was estimated by one-way ANOVA and Student's t-test, with Bonferroni-Holm method of p level correction.

The online version of this article includes the following source data for figure 5:

**Source data 1.** Endocytic and scaffold machineries co-operate for rapid vesicle replenishment during high-frequency transmission at the calyx of Held.

## Discussion

At the calyx of Held in slices under physiologically optimized conditions, blocking endocytosis by bath-application of Dynasore or Pitstop-2 enhanced activity dependent short-term depression of EPSCs evoked by high-frequency afferent fiber stimulations. This enhancement of synaptic depression was significant already at the 2nd EPSC (10 ms) after the stimulation onset and proceeded bi-exponentially with a fast (5–6 ms) and slow (24–35 ms) time constant (*Figure 5B*). These slow time constants were comparable to control time constant (37 ms) undergoing mono-exponential depression. Thus, the depression-counteracting effect of endocytosis was restricted to the early epoch of high-frequency transmission when many vesicle residues occupy the release sites. As number of vesicles undergoing exocytosis during repetitive stimulation gradually reduces, the site-clearance likely becomes less demanding.

At the calyx of Held, Dynasore or Pitstop-2 blocked both fast and slow endocytosis (*Figure 1*). The fast endocytosis is characterized with a rate of ~350 fF/s, which was slowed by endocytic blockers to ~150 fF/s (*Figure 1C*, *Supplementary file 1*). This reduction of endocytic rate (200 fF/s) corresponds to 2.54 vesicles/ms (1 fF = 12.7 vesicles; *Yamashita et al., 2010*; 200 fF/s x 12.7 vesicles x 1/1000ms). Since the mean amplitude of miniature EPSCs, representing a single vesicle response, was ~55 pA in 1.3 mM [Ca$^{2+}$] at PT (data not shown), endocytic block is estimated to reduce EPSC amplitude by ~1.4 nA in 10 ms (2.54 vesicles/ms x 55 pA x 10 ms x 1/1000 nA). This is close to the average amplitude difference between control and endocytic blockers (1.6 nA) at the 2nd EPSC (10 ms after the 1st EPSC) under 100 Hz stimulation (*Figure 2B1*, *Figure 2—figure supplement 1A*). Therefore, during high-frequency transmission, the site-clearance by fast endocytosis can fully compensate the initial strong depression. However, the rate of slow endocytosis (29 fF/s) is ~10 times slower than fast endocytosis (250–350 fF/s; *Figure 1A*, *Supplementary file 1*), therefore slow endocytosis unlikely contributes to the millisecond-order site-clearance. However, without slow endocytosis, the vesicular membrane remaining after spontaneous exocytosis would accumulate and congest release sites. Thus, slow endocytosis may play a house-keeping role in release site-clearance.

At the calyx of Held, scaffold protein inhibitors significantly enhanced the early phase of synaptic depression (starting at the 2nd stimulation at 100 Hz; *Figure 4B1*) like endocytic blockers (*Figure 2B1*; *Figure 5A and B*), despite that they do not block endocytosis (*Figure 3*). In contrast to the endocytic blockers (*Figure 2*), the STD-enhancing effect of scaffold protein inhibitors was stronger and independent of stimulation frequency (*Figure 4*). Co-application of endocytic and scaffold protein cascade inhibitors revealed that the synaptic strengthening effects of endocytosis and scaffold machinery are complementary rather than additive (*Figure 5C*). These results together suggest that fast endocytosis quickly clears vesicle residues, whereas scaffold machinery simultaneously mobilizes and replenishes transmitter-filled vesicles to release sites during high-frequency transmission (*Figure 7*). Through these combinatory mechanisms, synaptic strength and high-fidelity neurotransmission (up to 500 Hz; *Sonntag et al., 2009*) can be maintained at fast synapses such as the calyx of Held auditory relay synapse.

At hippocampal CA1 synapses the roles of endocytosis and scaffold machineries in fast vesicle replenishment were different from those at the calyx of Held. Under physiological conditions in slices, short-term plasticity at hippocampal CA1 synapses was facilitatory, in contrast to depression at calyceal synapses. Endocytic blockers attenuated the synaptic facilitation, suggesting that endocytosis boosts synaptic facilitation at this synapse (*Figure 6*). Since synaptic strength at facilitatory synapses is determined by a sum of facilitation and depression mechanisms, attenuation of synaptic

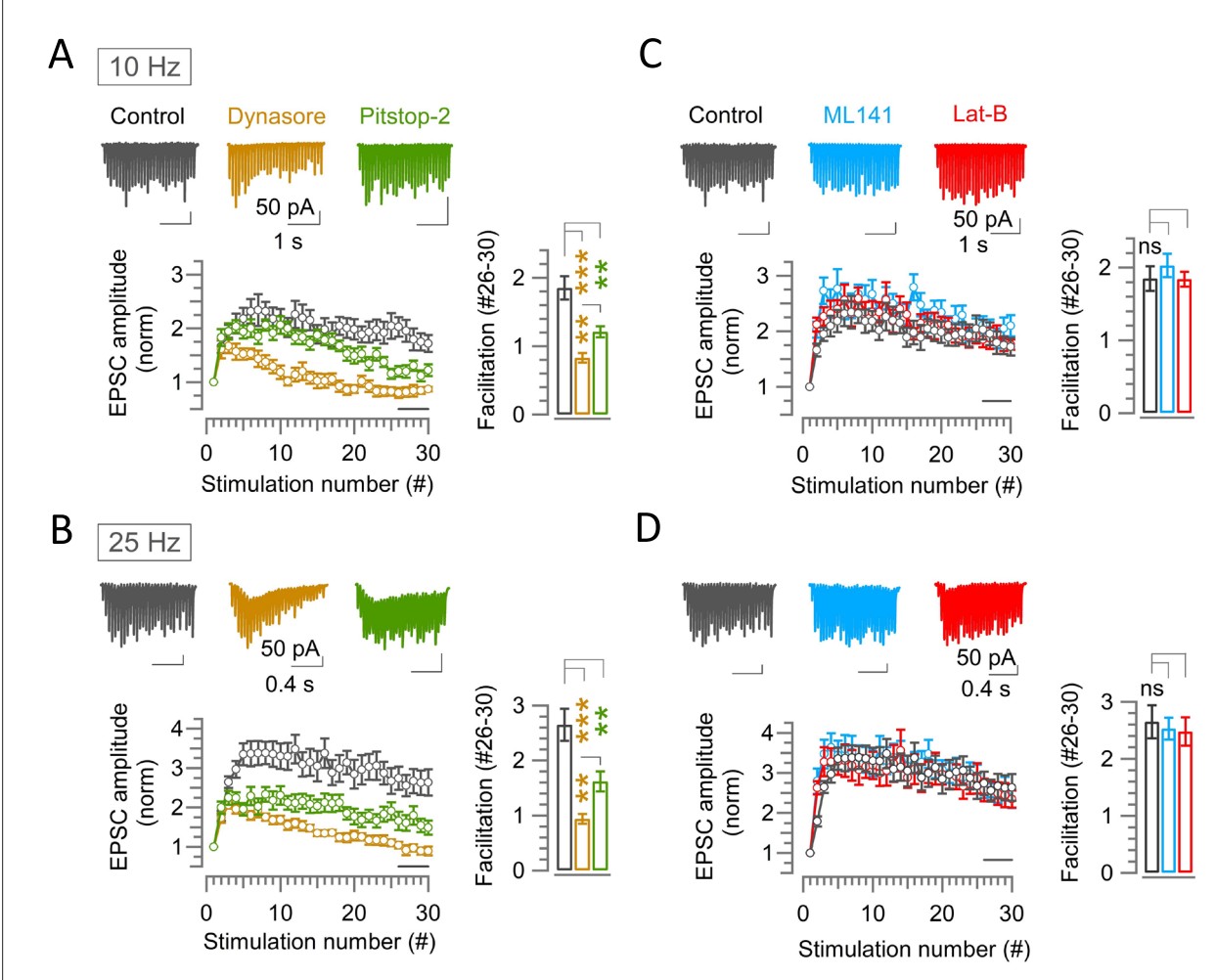

**Figure 6.** Endocytic blockers attenuate synaptic facilitation activity dependently at hippocampal CA1 synapses, but scaffold cascade inhibitors show no effect. (**A–D**) A train of 30 EPSCs recorded from hippocampal CA1 pyramidal cells evoked by Schaffer collaterals stimulations at 10 Hz (**A and C**) or 25 Hz (**B and D**) in the absence (control, black) or presence of endocytic blocker Dynasore (100 µM, 10–60 min, brown) or Pitstop-2 (25 µM, 10–60 min, green) (**A, B**), or scaffold protein inhibitor ML141 (10 µM, 10–60 min, cyan) or Latrunculin-B (15 µM, 10–60 min, red) (**C, D**) at PT (37°C) and in 1.3 mM Ca²⁺ aCSF. Top panels show sample EPSC traces. Lower panels show average EPSC amplitudes normalized and plotted against stimulation numbers. Bar graphs show EPSCs amplitudes averaged from #26–30 events. (**A**) At 10 Hz stimulation, EPSCs in control showed facilitation reaching a peak at the 7th stimulation (2.34±0.3; n=17). Thereafter, it gradually reduced. Dynasore significantly reduced the facilitation immediately before the peak at 6th stimulation (1.53±0.097; n=10; p=0.016) compared to control (2.32±0.22), but not Pitstop-2. Towards the end of stimulus train (#26–30), synaptic facilitation in control (1.85±0.17) was significantly attenuated by Dynasore (0.83±0.07; p<0.001) or Pitstop-2 (1.21±0.08; p=0.003). (**B**) At 25 Hz stimulation, synaptic facilitation peaked at the 12th stimulation in control (3.5±0.4; n=16), at which the facilitation was significantly attenuated by Dynasore (1.6±0.11; n=11; p<0.001, t-test) or Pitstop-2 (2.11±0.22; n=14; p=0.004). Also, at #26–30, synaptic facilitation in control (2.65±0.3) was strongly attenuated by Dynasore (0.94±0.1; p<0.001) or Pitstop-2 (1.62±0.2; p=0.007). (**C**) At 10 Hz stimulation, the peak facilitation at the 7th stimulation in control (2.34±0.3; n=17) was not significantly changed by ML141 (2.5±0.22; n=10; p=0.8, Student's t-test) or Lat-B (2.3±0.23; n=12; p=0.9). Likewise, the facilitation at #26–30 in control (1.85±0.17) was not altered by ML141 (2.03±0.16; p=0.5) or Lat-B (1.84±0.1; p=0.96). (**D**) At 25 Hz stimulation, peak facilitation at 12th stimulation in control (3.5±0.4; n=16) was unchanged by ML141 (3.3±0.34; n=10; p=0.72, t-test) or Lat-B (3.1±0.4; n=11; p=0.42). Facilitation at #26–30 events in control (2.65±0.3) was also unaltered by ML141 (2.53±0.2; p=0.8) or Lat-B (2.5±0.25; p=0.7). The significance of difference of all data in this figure was estimated by one-way ANOVA and Student's t-test, with Bonferroni-Holm method of p level correction.

The online version of this article includes the following source data and figure supplement(s) for figure 6:

**Source data 1.** Endocytic blockers attenuate synaptic facilitation activity dependently at hippocampal CA1 synapses, but scaffold cascade inhibitors show no effect.

**Figure supplement 1.** Vacuolar (v-) ATPase blockers, Bafilomycin or Folimycin does not affect the short-term synaptic facilitation at the hippocampal CA1 synapse.

**Figure supplement 1—source data 1.** Vacuolar (v-) ATPase blockers, Bafilomycin or Folimycin does not affect the short-term synaptic facilitation at the hippocampal CA1 synapse.

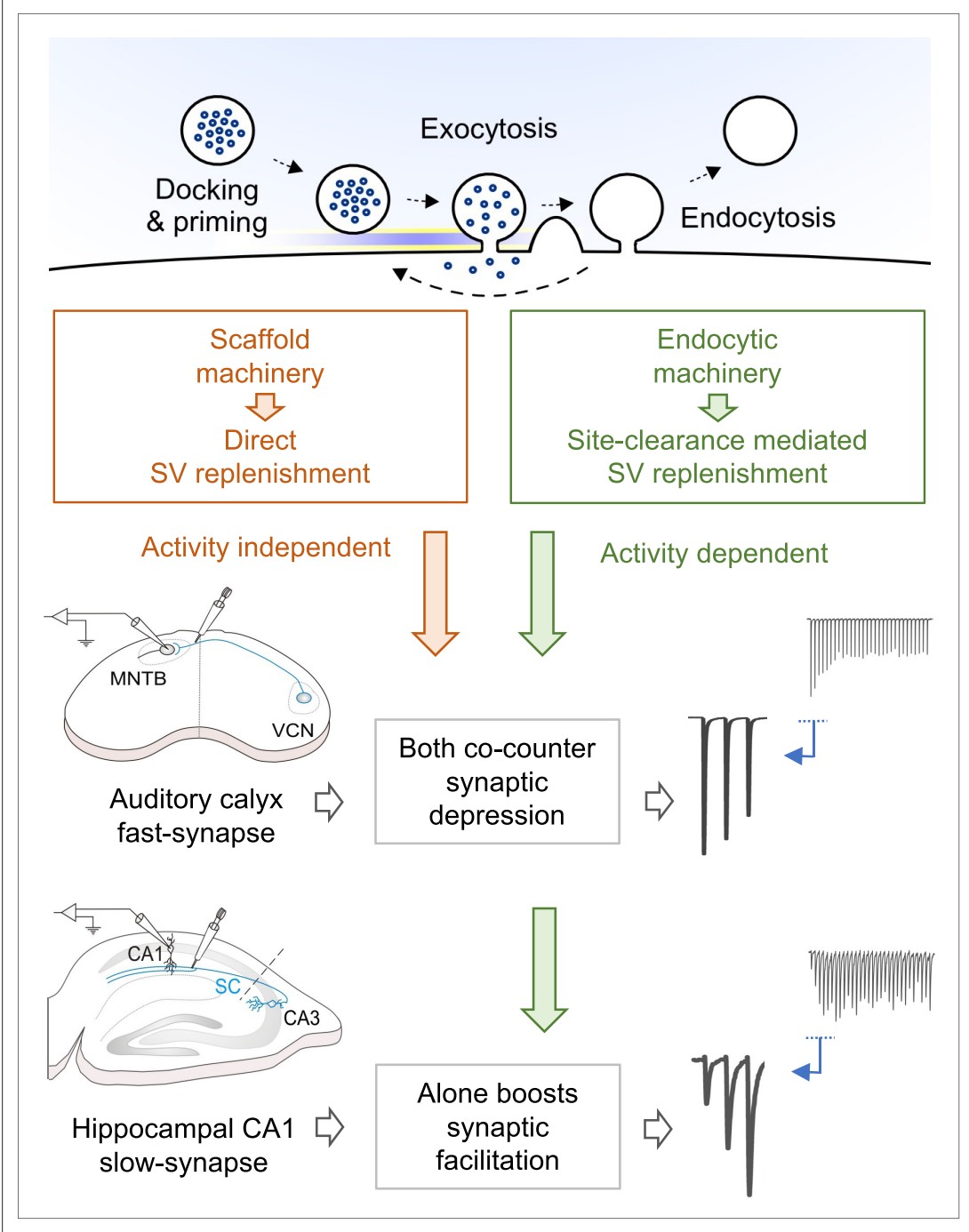

**Figure 7.** Hypothetical vesicle replenishment scheme by endocytosis and scaffold-machineries during repetitive transmission at fast calyx and slow hippocampal CA1 synapses. During high-frequency transmission, endocytosis driven site-clearance allows activity-dependent replenishment of new vesicles to release sites. This endocytic function counteracts synaptic depression caused by vesicle depletion, thereby maintain synaptic strength, enabling high-fidelity fast neurotransmission at sensory relay synapses, like at the calyx of Held. This endocytosis driven synaptic strengthening function augments synaptic facilitation at slow-plastic synapses like hippocampal CA1 synapses that exhibit long-term plasticity, thereby boosting its induction capability for memory formation. Whereas, the presynaptic scaffold machinery plays a powerful direct vesicle replenishment role, independent of endocytosis and activity, thereby rapidly translocating new vesicles to open release sites including those just opened by endocytic site-clearance. This scaffold function is specifically devoted to fast synapses, with high release probability but not to slow synapses, where vesicle depletion is minimal due to low release probability.

depression by endocytosis can boost synaptic facilitation. At hippocampal excitatory synapses, short-term increase in glutamate release induces long-term potentiation (LTP) of excitatory transmission. Thus, from physiological viewpoints, endocytosis-dependent site clearance may play a critical role in the induction of LTP underlying memory formation.

In clear contrast to fast calyceal synapses, the scaffold protein inhibitor ML141 or Latrunculin B had no effect on EPSC facilitation at relatively slow hippocampal CA1 synapses (*Figure 6C and D*). This was somewhat unexpected since in hippocampal culture Latrunculin-A blocks ultra-fast endocytosis (UFE, *Watanabe et al., 2013*), which occurs preferentially at physiological temperature (*Watanabe et al., 2014*). Since Latrunculin A and B are equipotent in enhancing STD at calyceal synapses in 1.3 mM [Ca $^{2+}$] aCSF at PT (*Figure 4—figure supplement 1C and D*), the physiological role of UFE in hippocampal synaptic transmission remains unclear. Latrunculin-B reportedly inhibits vesicular docking by limiting replacement pool filling during high-frequency inputs at parallel fiber (PF)-molecular layer interneuron synapses in cerebellar slices (*Miki et al., 2016*). This synapse is adapted to respond to high-frequency inputs, like other cerebellar synapses (e.g. *Ritzau-Jost et al., 2014*). Therefore, it is likely that the actin-dependent scaffold-machinery operates for rapid vesicular replenishment predominantly at fast synapses, whereas it is non-functional at relatively slow synapses. In contrast, the endocytosis-dependent site-clearance mechanism seems to be a universal phenomenon supporting vesicular replenishment among fast and slow mammalian central synapses.

# Materials and methods

**Key resources table**

| Reagent type (species) or resource | Designation | Source or reference | Identifiers | Additional information |
|---|---|---|---|---|
| Chemical compound, drug | Dynasore | Abcam | Cat# ab120192 | 100 µM |
| Chemical compound, drug | Pitstop-2 | Abcam | Cat# ab120687 | 25 µM |
| Chemical compound, drug | ML141 | Abcam | Cat# ab145603 | 10 µM |
| Chemical compound, drug | Latrunculin A | Labchem -Wako, Fujifilm | 125–04363 | 20 µM |
| Chemical compound, drug | Latrunculin B | Abcam | Cat# ab144291 | 15 µM |
| Chemical compound, drug | Folimycin (or Concanamycin A) | Abcam | Cat# ab144227 | 67 nM |
| Chemical compound, drug | Bafilomycin A1 | Cayman Chemical | Cat# 11038 | 5 µM |
| Peptide, recombinant protein | Dynamin-1 PRD peptide | GenScript | (Custom ordered) | 1 mM Sequence: PQVPSRPNRAP |
| Chemical compound, drug | Strychnine HCl | Sigma-Aldrich | Cat# S8753 | 2 µM |
| Chemical compound, drug | Bicuculline Methiodide | Sigma-Aldrich | Cat# 14343 | 10 µM |
| Chemical compound, drug | D-AP5 | Tocris | Cat# 0106 | 50 µM |
| Chemical compound, drug | QX-314 bromide | Alomone Labs | Cat# Q-100 | 2 mM |
| Chemical compound, drug | DMSO, Sterile Filtered | Santa Cruz Chemicals | Cat# sc-359032 | See in Methods details |
| Other | EPC 10 USB Patch Clamp Amplifier | Heka Elektronik | N/A | See in Method details |
| Other | BX51WI upright microscope | Olympus | Cat# BX51WI | See in Method details |
| Other | VT1200S Vibratome | Leica | N/A | See in Method details |
| Other | Model 2100 Isolated Pulse Stimulator | A-M Systems | N/A | See in Method details |
| Other | TC-344C Dual Channel Temp Controller | Warner Instruments | TC-344C | See in Method details |

*Continued on next page*

*Continued*

| Reagent type (species) or resource | Designation | Source or reference | Identifiers | Additional information |
|---|---|---|---|---|
| Other | PatchStar Micromanipulator | Scientifica | PatchStar | For precise movement and positioning of recording probe and stimulation electrode |
| Other | Axiocam 506 mono camera system | Zeiss | Axiocam 506 mono | Visualization system for slice recording |
| Other | PIP 6 pipette puller | HEKA | PIP 6 | See in Method details |
| Other | Borosilicate glass capillary | King Precision Glass Inc. | N/A | (2.0 mm OD) |
| Software, algorithm | Patchmaster | HEKA | N/A | |
| Software, algorithm | ZEN 2 Lite | Zeiss | N/A | |
| Software, algorithm | IGOR Pro | Wavemetrics | N/A | |
| Software, algorithm | Microsoft Excel | Microsoft | N/A | |
| Software, algorithm | Corel Draw | Corel Corporation | N/A | |

## Animals

All experiments were performed in accordance with the guidelines of the Physiological Society of Japan and animal experiment regulations at Okinawa Institute of Science and Technology Graduate University. C57BL/6J mice of either sex after hearing onset (postnatal day 13–15) were used for the experiment and animals were maintained on a 12 hr light/dark cycle with food and water ad libitum.

## Method details

### Slice preparation (brainstem and hippocampal) and electrophysiology

Following decapitation of C57BL/6J mice under isoflurane anesthesia, brains were isolated and transverse slices (200 µm thick) containing the medial nucleus of the trapezoid body (MNTB) or parasagittal slices (300 µm) containing the hippocampus were cut using a vibratome (VT1200S, Leica) in ice-cold artificial cerebrospinal fluid (aCSF, see below) with reduced $Ca^{2+}$ (0.1 mM) and increased $Mg^{2+}$ (2.9 mM) concentrations and with NaCl replaced by 200 mM sucrose. Slices were incubated for 1 hr at 37°C in standard aCSF containing (in mM); 115 NaCl, 2.5 KCl, 25 $NaHCO_3$, 1.25 $NaH_2PO_4$, 1.3 or 2.0 $CaCl_2$, 1 $MgCl_2$, 10 glucose, 3 myo-inositol, 2 sodium pyruvate, and 0.4 sodium ascorbate (pH 7.3–7.4 when bubbled with 95% $O_2$ and 5% $CO_2$, 285–290 mOsm). Unless otherwise noted, EPSCs were recorded in 1.3 mM $[Ca^{2+}]$ aCSF at 37°C. During recording, a brain slice in a temperature-controlled chamber (RC-26GLP, PM-1; Warner instruments) was continuously perfused with aCSF solutions without (control) or with the pharmacological blockers using a peristaltic pump (5–6 µl/s). Recordings with or without the blockers were made from separate cells in different slices from a minimum of 3–4 animals for each set of experiment. In experiments at PT, solutions kept in a water bath (37°C) and passed through an in-line heater (SH-27B, Warner instruments) placed immediately before the recording chamber and maintained at 37°C (±0.2°C) by a controller (TC-344C, Warner instruments). For RT experiments, no heating was used, and the temperature was within 22–24°C. Recordings in the presence of pharmacological blockers were made within 10–60 min of drug application and for control experiments a slice was kept in the recording chamber no longer than 1 hr.

Whole-cell recordings were made using a patch-clamp amplifier (EPC 10 USB, HEKA Elektronik, Germany) from the calyx of Held presynaptic terminals, postsynaptic MNTB principal neurons, or hippocampal CA1 pyramidal neurons visually identified with a 60X water immersion objective (LUMPlanFL, Olympus) attached to an upright microscope (BX51WI, Olympus, Japan). Data were acquired at a sampling rate of 50 kHz using Patchmaster software (for EPC 10 USB) after online filtering at 5 kHz. The patch pipettes were pulled using a vertical puller (PIP6, HEKA) and had a resistance of 3–4 MΩ.

### Membrane capacitance recordings from presynaptic calyx terminal (at PT)

The patch pipettes for presynaptic capacitance recording were wax coated to reduce stray capacitance and had a series resistance of 6–15 MΩ, which was compensated by up to 60% for its final value

to be 6 MΩ. To isolate presynaptic $Ca^{2+}$ currents ($I_{Ca}$) during membrane capacitance measurements ($C_m$), tetrodotoxin (1 µM) and tetra-ethyl-ammonium chloride (TEA, 5 mM) were routinely added to block $Na^+$ and $K^+$ channels, respectively. Calyx terminals were voltage clamped at a holding potential of –70 mV and a sine wave (peak to peak = 60 mV at 1 kHz) was applied. A single pulse (5 ms) or a train of 20 ms pulses (15 stimuli at 1 Hz or 10 stimuli at 10 Hz) to +10 mV were given to induce slow, fast-accelerating, and fast endocytosis. The capacitance trace within 450 ms after the stimulation pulse was excluded from analysis to avoid capacitance artifacts (*Yamashita et al., 2005*). The exocytic capacitance jump ($\Delta C_m$) in response to a stimulation pulse was measured as difference between baseline and mean $C_m$ at 450–460 ms after the stimulation pulse. The endocytic rate was measured from the $C_m$ decay phase 0.45–5.45 s from the end of 5 ms stimulation pulse, 0.45–0.95 s for 1 Hz train stimulation and 0.45–1.45 s after the last (10th) pulse for 10 Hz train stimulation. The average $C_m$ traces were calculated from individual $C_m$ traces, and error bars were analyzed from resampled individual $C_m$ traces in low frequency for clarity (every 10 ms for 10 Hz train stimulation and every 20 ms for both 5 ms single and 1 Hz train stimulations).

### EPSC recordings from brainstem and hippocampus

For EPSC recordings, brainstem MNTB principal neurons or hippocampal CA1 pyramidal cells were voltage-clamped at the holding potential of –80 mV. To evoke EPSCs, a bipolar electrode was placed on afferent fibers between MNTB and midline in brainstem slices or on the Schaffer collateral fibers in the stratum radiatum near the CA1 border in hippocampal slices.

The patch pipettes had a series resistance of 5–15 MΩ (less than 10 MΩ in most cells) that was compensated up to 80% for a final value of ~3 MW for EPSC recording from MNTB neurons. For EPSC recording from hippocampal pyramidal cells, no compensation was made, and capacitance artifacts were manually corrected. Stimulation pulses were applied through an isolator (Model 2100, A-M Systems) controlled by EPC10 amplifier (HEKA).

During EPSC recordings, strychnine-HCl (Sigma, 2 µM), bicuculline-methiodide (Sigma, 10 µM) and D-(-)–2-Amino-5-phosphonopentanoic acid (D-AP5, Tocris, 50 µM) were added to isolate AMPA receptor-mediated EPSCs. In some experiments, to minimize AMPA receptor saturation and desensitization, kynurenic acid (Sigma, 1 mM) was added to aCSF. Unless otherwise specified, the internal pipette solution for presynaptic capacitance measurements contained (in mM): 125 Cs gluconate, 10 HEPES, 20 TEA-Cl, 5 sodium phosphocreatine, 4 Mg-ATP, 0.3 Na-GTP and 0.5 EGTA (pH 7.2–7.3, adjusted with CsOH, adjusted to 305–315 mOsm by varying Cs-gluconate concentration). For EPSC recording, the internal pipette solution contained QX-314 bromide (2 mM) and EGTA, either 5 mM (for MNTB cells) or 0.5 mM (for pyramidal cells), but otherwise identical to the presynaptic pipette solution. Dynasore (100 µM), Pitstop-2 (25 µM), ML 141 (10 µM), Folimycin (also known as Concanamycin A; 67 nM), Latrunculin-A (20 µM), Latrunculin B (15 µM), and Bafilomycin A1 (5 µM) were dissolved in DMSO (final concentration 0.1%, except for Latrunculin-A, 0.2%). Since Latrunculin A, Latrunculin-B and Folimycin are light sensitive, precautions were taken to minimize light exposure during handling and recording. Dynamin 1 proline-rich domain (PRD) peptide (sequence: PQVPSRPNRAP; GenScript) was dissolved in presynaptic pipette solution (1 mM). Membrane capacitance ($C_m$) recordings in the presence of Dyn-1 PRD peptide were initiated 4–5 min after whole-cell membrane rupture to allow its time of diffusion inside presynaptic terminals.

## Data analysis and statistics

Experiments were designed as population study using different cells from separate brain slices under control and drug treatment, rather than on a same cell before and after the drug exposure. All data were analyzed using IGOR Pro (WaveMetrics) and Microsoft Excel. All values are given as mean ± SEM. The significance of effects of multiple drug applications were determined by one-way ANOVA, keeping the significance level set at $p<0.05$, denoted with asterisks (*$p<0.05$, **$p<0.01$, ***$p<0.001$). Once found to be significantly different, each drug effect was compared with the control using Student's t-test with Bonferroni-Holm method of post-hoc test to correct the p level.

## Acknowledgements

We thank Yukiko Goda for comments, Patrick Stoney for editing this paper. This research was supported by funding from Okinawa Institute of Science and Technology to TT, and Grants-in-Aid for Scientific Research from the Japan Society for the Promotion of Science (21K06445) to SM.

## Additional information

### Funding

| Funder | Grant reference number | Author |
| --- | --- | --- |
| Okinawa Institute of Science and Technology Graduate University | | Tomoyuki Takahashi |
| Japan Society for the Promotion of Science | 21K06445 | Satyajit Mahapatra |

The funders had no role in study design, data collection and interpretation, or the decision to submit the work for publication.

### Author contributions

Satyajit Mahapatra, Conceptualization, Data curation, Software, Formal analysis, Funding acquisition, Validation, Investigation, Visualization, Methodology, Writing - original draft, Project administration, Writing - review and editing; Tomoyuki Takahashi, Conceptualization, Resources, Software, Supervision, Funding acquisition, Validation, Visualization, Methodology, Writing - original draft, Project administration, Writing - review and editing

### Author ORCIDs

Satyajit Mahapatra (ID) http://orcid.org/0000-0002-4370-218X
Tomoyuki Takahashi (ID) http://orcid.org/0000-0002-8771-7666

### Ethics

All experiments were performed in accordance with the guidelines of the Physiological Society of Japan and animal experiment regulations at Okinawa Institute of Science and Technology Graduate University.

Joint Public Review: https://doi.org/10.7554/eLife.90497.4.sa1
Author response https://doi.org/10.7554/eLife.90497.4.sa2

## Additional files

### Supplementary files

• Supplementary file 1. Membrane capacitance and calcium current amplitudes with or without endocytic blockers or presynaptic scaffold inhibitors during slow, fast-accelerating, or fast endocytosis.

• Supplementary file 2. Parameters of recovery from STD at 37°C and 1.3 mM $Ca^{2+}$ with or without endocytic- or presynaptic scaffold-blockers.

• MDAR checklist

### Data availability

All source data for main and supplementary figures contain the numerical data used to generate the figures. Further information about resources should be addressed to authors.

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
