## [Editor Report · eLife assessment]

Following synaptic vesicle fusion events at release sites, vesicle remnants will need to be cleared in order to allow new rounds of vesicle docking and fusion. This **fundamental** study of Mahapatra and Takahashi examines the role of release site clearance in synaptic transmission during repetitive activity in two types of central synapses, the giant calyx of Held and hippocampal CA1 synapses. The study uses pharmacological approaches to interfere with release site clearance by blocking membrane retrieval (endocytosis). The results also show how pharmacological inhibition of scaffold proteins affects short-term plasticity. The data presented make a **compelling** case for fast endocytosis as necessary for rapid site clearance and vesicle recruitment to active zones. The data reveal an unexpected, fast role for local site clearance in counteracting synaptic depression.

---

## [Referee Report · Joint Public Review]

Mahapatra and Takahashi report on the physiological consequences of pharmacologically blocking either clathrin and dynamin function during compensatory endocytosis or of the cortical actin scaffold both in the calyx of Held synapse and hippocampal boutons in acute slice preparations.

Although many aspects of these pharmacological interventions have been studied in detail during the past decades, this is a comprehensive and comparative study, which reveals some interesting differences between a fast synapse (Calyx of Held) tuned to reliably transmit at several 100 Hz and a more slow hippocampal CA1 synapse. In particular the authors find that acute disturbance of the synaptic actin network leads to a marked frequency-dependent enhancement of synaptic depression in the Calyx, but not in the hippocampal synapse. This striking difference between both preparations is the most interesting finding.

Comments on latest version:

The authors have done a great job revising the paper and only minor revisions are suggested to the Discussion of the paper.

Two quite relevant and recent papers should be cited and briefly discussed because they relate directly to Pitstop2 effects and actin-myosin-scaffold proteins in the calyx of Held synapse.

One is: Paksoy A et al, (2022) "Effects of the clathrin inhibitor Pitstop-2 on synaptic vesicle recycling at a central synapse in vivo." Front. Synaptic Neurosci. 14:1056308. doi: 10.3389/fnsyn.2022.1056308. This paper shows with EM that changes caused by PitStop2 perturbation of "clathrin function suggest that clathrin plays a role in SV recycling from both, the plasma membrane and large endosomes, under physiological activity patterns, in vivo."

Second: A role for actin-myosin and MLCK in short-term plasticity has been shown by Srinivasan G., et al. (2008) "The Pool of Fast Releasing Vesicles Is Augmented by Myosin Light Chain Kinase Inhibition at the Calyx of Held Synapse." J Neurophysiol 99: 1810-1824, 2008. The data here suggests that MLCK plays a crucial role in determining the size of the pool of synaptic vesicles that undergo fast release but not the Pr of the synapse. In other words, MLCK inhibition augments super-priming of vesicles at the calyx of Held synapse.

---

## [Author Response]

The following is the authors’ response to the previous reviews.

**eLife assessment**
Following synaptic vesicle fusion events at release sites, vesicle remnants will need to be cleared in order to allow new rounds of vesicle docking and fusion. This fundamental study of Mahapatra and Takahashi examines the role of release site clearance in synaptic transmission during repetitive activity in two types of central synapses, the giant calyx of Held and hippocampal CA1 synapses. The study uses pharmacological approaches to interfere with release site clearance by blocking membrane retrieval (endocytosis). They compare the effects on short-term plasticity with those obtained by pharmacologically inhibiting scaffold protein activity. The data presented make a compelling case for fast endocytosis as necessary for rapid site clearance and vesicle recruitment to active zones. The data reveal an unexpected, fast role for local site clearance in counteracting synaptic depression.
**Public Reviews:**

**Reviewer #1 (Public Review):**
Summary:The study examines the role of release site clearance in synaptic transmission during repetitive activity under physiological conditions in two types of central synapses, calyx of Held and hippocampal CA1 synapses. After acute block of endocytosis by pharmacology, deeper synaptic depression or less facilitation was observed in two types of synapses. Acute block of CDC42 and actin polymerization, which possibly inhibits the activity of Intersectin, affected synaptic depression at the calyx synapse, but not at CA1 synapses. The data suggest an unexpected, fast role of the site clearance in counteracting synaptic depression.Strengths:The study uses acute block of the molecular targets with pharmacology together with precise electrophysiology. The experimental results are clear cut and convincing. The study also examines the physiological roles of the site clearance using action potential-evoked transmission at physiological Ca and physiological temperature at mature animals. This condition has not been examined.Weaknesses:Pharmacology may have some off-target effects, though acute manipulation should be appreciated and the authors have tried several reagents to verify the overall conclusions.
**Reviewer #2 (Public Review):**
Summary:In this manuscript, Mahapatra and Takahashi report on the physiological consequences of pharmacologically blocking either clathrin and dynamin function during compensatory endocytosis or of the cortical actin scaffold both in the calyx of Held synapse and hippocampal boutons in acute slice preparationsStrengths:Although many aspects of these pharmacological interventions have been studied in detail during the past decades, this is a nice comprehensive and comparative study, which reveals some interesting differences between a fast synapse (Calyx of Held) tuned to reliably transmit at several 100 Hz and a more slow hippocampal CA1 synapse. In particular the authors find that acute disturbance of the synaptic actin network leads to a marked frequency-dependent enhancement of synaptic depression in the Calyx, but not in the hippocampal synapse This striking difference between both preparations is the most interesting and novel finding.Weaknesses:Unfortunately, however, these findings concerning the different consequences of actin depolymerization are not sufficiently discussed in comparison to the literature. My only criticism concerns the interpretation of the ML 141 and Lat B data. With respect to the Calyx data, I am missing a detailed discussion of the effects observed here in light of the different RRP subpools SRP and FRP. This is very important since Lee at al. (2012, PNAS 109 (13) E765-E774) showed earlier that disruption of actin inhibits the rapid transition of SRP SVs to the FRP at the AZ. The whole literature on this important concept is missing. Likewise, the role of actin for the replacement pool at a cerebellar synapse (Miki et al., 2016) is only mentioned in half a sentence. There is quite some evidence that actin is important both at the AZ (SRP to FRP transition, activation of replacement pool) and at the peri-active zone for compensatory endocytosis and release site clearance. Both possible underlying mechanisms (SRP to FRP transition or release site clearance) should be better dissected.

We dissected the latrunculin effect further by referring to the related literature within the scope of this study in the revised Discussion section (last paragraph).

**Reviewer #3 (Public Review):**
The manuscript by Mahapatra and Takahashi addresses the role of presynaptic release site clearance during sustained synaptic activity. The authors characterize the effects of pharmacologically interfering with SV endocytosis (pre-incubation with Dynasore or Pitstop-2) on synaptic short-term plasticity (STP) at two different CNS synapses (calyx of Held synapses and hippocampal SC to CA1 synapses) using patch-clamp recordings in acute slices under experimental conditions designed to closely mimic a physiological situation (37{degree sign}C and 1.3 mM external [Ca2+]). Endocytosis blocker-induced changes in STP and in the recovery from short-term depression (STD) are compared to those seen after pharmacologically inhibiting actin filament assembly (pre-incubation with Latrunculin-B or the selective Cdc42 GTPase inhibitor ML-141). Presynaptic capacitance (Cm) recordings in calyx terminals were used to establish the effects of the pharmacological maneuvers on SV endocytosis.Latrunculin-B and ML-141 neither affect SV endocytosis (assayed by Cm recordings) nor EPSC recovery following conditioning trains, but strongly enhances STD at calyx synapses. No changes in STP were observed at Latrunculin-B- or ML-141-treated SC to CA1 synapses.Dynasore and Pitstop-2 slow down endocytosis, limit the total amount of exocytosis in response to long stimuli, enhance STD in response to 100 Hz stimulation, but profoundly accelerate EPSC recovery following conditioning 100 Hz trains at calyx synapses. At SC to CA1 synapses, Dynasore and Pitstop-2 reduce the extend of facilitation and lower relative steady-state EPSCs suggesting a change in the facilitation-depression balance in favor of the latter.The authors use state-of-the art techniques and their data, which is clearly presented, leads to authors to conclude that endocytosis is universally important for clearance of release sites while the importance of scaffold protein-mediated site clearance is limited to 'fast synapses'.Unfortunately, and perhaps not completely unexpected in view of the pharmacological tools chosen, there are several observations which remain difficult to understand:(1) Blocking site clearance affects release sites that have previously been used, i.e. sites at which SV fusion has occurred and which therefore need to be cleared. Calyces use at most 20% of all release sites during a single AP, likely fewer at 1.3 mM external [Ca2+]. Even if all those 20% of release sites become completely unavailable due to a block of release site clearance, the 2nd EPSC in a train should not be reduced by >20% because ~80% of the sites cannot be affected. However,~50% EPSC reduction was observed (Fig. 2B1, lower right panel) raising the possibility thatDynasore does more than specifically interfering with SVs endocytosis (and possibly Pitstop as well). Non-specific effects are also suggested by the observed two-fold increase in initial EPSC size in SC to CA1 synapses after Dynasore pre-incubation.

This study compares different experimental conditions to conclude the physiological role of endocytosis on rapid neurotransmission at the large calyceal synapse in mice. A related study at the *Drosophila* neuromuscular junction (Kawasaki et al., Nat. Neuroscience 2000) reported similar findings in comparable experimental settings (physiological conditions and acute block of endocytosis).

(2) More severe depression was observed at calyx synapses after blocking endocytosis which the authors attribute to a presynaptic mechanism affecting pool replenishment. When probing EPSC recovery after conditioning 100 Hz trains, a speed up was observed mediated by an "unknown mechanism" which is "masked in 2 mM [Ca2+]". These two observations, deeper synaptic depression during 100 Hz but faster recovery from depression following 100 Hz, are difficult to align and no attempt was made to find an explanation.

By varying temperature (PT vs RT), calcium concentration (1.3 mM vs 2.0 mM), and stimulation frequency (10, 100, and 200 Hz; some data are not shown), the effect of endocytosis block on EPSC STD and recovery from STD kinetics at the post-hearing calyx were compared in these settings: (PT, 1.3 mM [Ca2+]), (PT, 2.0 mM Ca2+), and (RT, 2.0 mM [Ca2+]), to dissect their respective role.

(3) To reconcile previous data reporting a block of Ca2+-dependent recovery (CDR) by Dynasore or Latrunculin (measured at 2 mM external [Ca2+]) with the data presented here (using 1.3 mM external [Ca2+]) reporting no effect or a speed up of recovery from depression, the authors postulate that "CDR may operate only when excessive Ca2+ enters during massive presynaptic activation" (page 10 line 244). While that is possible, such explanation ignores plenty of calyx studies demonstrating fiber stimulation-induced CDR and elucidating molecular pathways mediating fiber stimulation-induced CDR, and it also completely dismisses the strong change in recovery time course after 10 Hz conditioning (single exponential) as compared to 100 Hz conditioning (double exponential with a pronounced fast component).Strong presynaptic stimuli such as those illustrated in Figs. 1B,C induce massive exocytosis. The illustrated Cm increase of 2 to 2.5 pF represents fusion of 25,000 to 30,000 SVs (assuming a single SV capacitance of 80 aF) corresponding to a 12 to 15% increase in whole terminal membrane surface (assuming a mean terminal capacitance of ~16 pF). Capacitance measurements can only be considered reliable in the absence of marked changes in series and membrane conductance. Documentation of the corresponding conductance traces is therefore advisable for such massive Cm jumps and merely mentioning that the first 450 ms after stimulation were skipped during analysis or referring to previous publications showing conductance traces is insufficient.All bar graphs in Figures 1 through 6 and Figures S3 through S6 compare three or even four (Fig. 5C) conditions, i.e. one control and at least two treatment data sets. It appears as if repeated t-tests were used to run multiple two-group comparisons (i.e. using the same control data twice for two different comparisons). Either a proper multiple comparison test should be used or a Bonferroni correction or similar multiple-comparison correction needs to be applied.

We updated the statistical analysis of all data using one-way ANOVA and t-test with Bonferroni-Holm method of p level correction and rectified one analysis in Fig 1 and 3, all major conclusions are unchanged.

Finally, the terminology of contrasting "fast-signaling" (calyx synapses) and "slow-plastic" (SC synapses) synapses seems to imply that calyx synapses lack plasticity, as does the wording "conventional bouton-type synapses involved in synaptic plasticity" (page 11, line 251). I assume, the authors primarily refer to the maximum frequencies these two synapse types typically transmit (fast-signaling vs slow-signaling)?

Properties of these two synapses described explicitly in updated text and they are renamed as fast and slow synapes.

**Recommendations for the authors:**

**Reviewer #3 (Recommendations For The Authors):**
'SV replenishment' and 'site clearance' should not be used synonymously as it seems to be done sometimes here.

In this revision, we described them more explicitly.

The data presented in Fig. S6 are detached from the rest of the manuscript, not relevant and should be removed. page 4 line 95 "... to ensure sufficient Ca2+ currents to induce exo-endocytosis." ICa is large enough to induce exocytosis also at 1.3 mM Ca2+. Please clarify.

We updated the relevant section.

page 5, line 108 "... this slow endocytosis showed a strongly prolonged time course without accompanied by the change of Cm or presynaptic Ca2+ currents" Please fix.

Fixed.

page 5, line 121 "Thus, at calyces of Held, bath-application of Dynasore or Pitstop-2 can block both fast and slow endocytosis without perturbing presynaptic intracellular milieu." Bath-application never perturbs the intracellular milieu. Please clarify.

Rephrased.

page 6 line 128 "... physiological aCSF" is a misnomer (=physiological artificial CSF). Please fix.

In the introduction section, it is clearly described.

page 11, line 252 "... from hippocampal SC-CA1 pyramidal neurons" There are no "SC-CA1 pyramidal neurons". Please fix.

Fixed.

page 12, line 285 "In acute slices optimized to physiological conditions" The conditions are optimized, not the slices. Please fix.

Fixed.

page 14, line 323 same as above

Fixed.

page 14, line 330 LTP at SC-CA1 synapses is postsynaptic. Please clarify.

Rephrased

page 16, line 381 "had a series resistance of 3-4 MOhm" versuspage 17, line 408 "The patch pipettes had a series resistance of 5-15 MOhm (less than 10 MOhm in most cells)" 3-4 is perhaps pipette resistance while 5-15 is perhaps series resistance? Please clarify.

Fixed.

page 17, line 398 "Cm traces were averaged at every 10 ms (for 10 Hz train stimulation) or 20 ms (for 5 ms single or 1 Hz train stimulation)." Do you mean to say that Cm traces were smoothed with a moving average using a window size of 10 or 20 ms duration? Please clarify.

Rephrased to clarify better.

page 18, "All values are given as mean {plus minus} SEM and significance of difference was evaluated by Student's unpaired t-test, unless otherwise noted." Please check. You cannot simply use repeated t-tests for multiple comparisons. Either a proper multiple comparison test should be used or a Bonferroni correction or similar multiple-comparison correction needs to be applied.

All statistical analysis are updated using one-way ANOVA and t-test, with Bonferroni-Holm method of p level correction and one analysis is rectified in Fig 1 and 3, with no change in major conclusions.